# Agent-based model predicts that layered structure and 3D movement work synergistically to reduce bacterial load in 3D in vitro models of tuberculosis granuloma

**Alexa Petrucciani[1], Alexis Hoerter[1], Leigh Kotze[2], Nelita Du Plessis[2], Elsje Pienaar [1,3]***

**1** Weldon School of Biomedical Engineering, Purdue University, West Lafayette, Indiana, United States of America, **2** DSI-NRF Centre of Excellence for Biomedical Tuberculosis Research, South African Medical Research Council for Tuberculosis Research, Division of Molecular Biology and Human Genetics, Faculty of Medical and Health Sciences, Stellenbosch University, Cape Town, South Africa, **3** Regenstrief Center for Healthcare Engineering, Purdue University, West Lafayette, Indiana, United States of America

* epienaar@purdue.edu

**Data Availability Statement:** We can change these references from doi numbers to full links as below. "All output data from our model along with code

## Abstract

Tuberculosis (TB) remains a global public health threat. Understanding the dynamics of host-pathogen interactions within TB granulomas will assist in identifying what leads to the successful elimination of infection. *In vitro* TB models provide a controllable environment to study these granuloma dynamics. Previously we developed a biomimetic 3D spheroid granuloma model that controls bacteria better than a traditional monolayer culture counterpart. We used agent-based simulations to predict the mechanistic reason for this difference. Our calibrated simulations were able to predict heterogeneous bacterial dynamics that are consistent with experimental data. In one group of simulations, spheroids are found to have higher macrophage activation than their traditional counterparts, leading to better bacterial control. This higher macrophage activation in the spheroids was not due to higher counts of activated T cells, instead fewer activated T cells were able to activate more macrophages due to the proximity of these cells to each other within the spheroid. In a second group of simulations, spheroids again have more macrophage activation but also more T cell activation, specifically CD8+ T cells. This higher level of CD8+ T cell activation is predicted to be due to the proximity of these cells to the cells that activate them. Multiple mechanisms of control were predicted. Simulations removing individual mechanisms show that one group of simulations has a CD4+ T cell dominant response, while the other has a mixed/CD8+ T cell dominant response. Lastly, we demonstrated that in spheroids the initial structure and movement rules work synergistically to reduce bacterial load. These findings provide valuable insights into how the structural complexity of *in vitro* models impacts immune responses. Moreover, our study has implications for engineering more physiologically relevant *in vitro* models and advancing our understanding of TB pathogenesis and potential therapeutic interventions.

for processing and visualization is archived on Zenodo (https://zenodo.org/doi/10.5281/zenodo.10035530 and https://zenodo.org/doi/10.5281/zenodo.10035532). The model code can be found at https://github.itap.purdue.edu/ElsjePienaarGroup/TB-in-vitro release v1.0.1.

**Funding:** This work used the Extreme Science and Engineering Discovery Environment (XSEDE). XSEDE resources from Anvil at Purdue and Expanse at UCSD were used through allocation TG-MDE220002. XSEDE is supported by National Science Foundation grant number ACI-1548562. The work of NdP is made possible through funding by the National Institute of Allergy and Infectious Diseases (NIAID), NIH, through contract #75N93019C0070. The work of AH is made possible in part by Grant TL1TR002531 (T. Hurley, PI) from the National Institutes of Health, National Center for Advancing Translational Sciences, Clinical and Translational Sciences Award. The funders had no role in study design, data collection and analysis, decision to publish, or preparation of the manuscript.

**Competing interests:** The authors have declared that no competing interests exist.

## Author summary

Tuberculosis (TB) is a respiratory infection that is associated with granulomas, organized clusters of immune cells that form around the bacteria. Recent work has focused on making *in vitro* models that better approximate these granulomas, such as our spheroid granuloma model. We found that this spheroid was better able to control bacteria than its traditional culture counterpart, so, in this work, we use agent-based modeling to explore the causes of this differential control. The simulated spheroids have increased macrophage activation. Surprisingly, this is not due to larger numbers of activated T cells, which activate these macrophages. Rather, the increase in activated macrophages is due to their proximity to fewer activated T cells. We also see a subset of simulations where the spheroids have increased cytotoxic CD8+ T cell activation. The simulation is further used to simulate changes in the structure of the *in vitro* models to explore its impact on immune response and bacterial control. These findings reveal the importance of structural complexity in *in vitro* models and offer insights into TB pathogenesis. Ultimately, our work paves the way for more realistic *in vitro* models and a deeper understanding of how our bodies combat TB.

## 1. Introduction

Before Covid-19, tuberculosis (TB) was the leading cause of death by a single infectious disease [1]. While much progress has been made in reducing TB mortality in the past decades, the Covid-19 pandemic has reversed some of this progress. In 2021, there were an estimated 1.6 million TB deaths, returning to the levels around 2017 [1]. An additional 10.6 million people fell ill with TB in 2021, which represents an increase in incidence that has not been seen in decades [1]. Thus, TB continues to be a global challenge.

TB is caused by *Mycobacterium tuberculosis (Mtb)*, which is transmitted through the air by respiratory droplets. When inhaled into the lungs these bacteria can lead to the formation of granulomas. Granulomas are organized structures of immune cells that both contain and shelter the bacteria. Granulomas typically have an interior core of myeloid cells such as macrophages and an exterior cuff of lymphocytes such as T cells. Previous computational modeling has predicted that the spatial organization of granulomas impacts T cell activation and bacterial control. [2] Understanding host-pathogen dynamics within granulomas, as well as the role of the granuloma structure, will be key to identifying protective host responses.

TB granulomas can be experimentally investigated in many ways. Tissue excised from TB patients can be analyzed to determine the spatial organization of specific cells and molecules in mature granulomas [3]. However, these samples typically represent more advanced disease as the samples are often obtained from lung resection surgeries or autopsies. Animal models such as non-human primates, rabbits, guinea pigs, zebrafish, and mice can provide insights into granuloma dynamics earlier in infection, as well as into the immune response at the whole organism level [4]. Genetic or pharmacologic manipulations of key molecules and pathways can be induced to further assess host-pathogen interactions and their impacts [4]. *In vitro* cell cultures can be used to reproduce specific features of granulomas in a more controlled environment. These are composed of various combinations of immune cells (e.g., monocytes, macrophages, PBMCs), non-immune cells (i.e., fibroblast and epithelial cells), extracellular matrix, and bacteria [5].

*In vitro* TB models can be subdivided into "2D" and "3D" cultures. 2D cultures are typically less complex, less expensive, and more reproducible [6,7]. However, 2D cultures do not fully

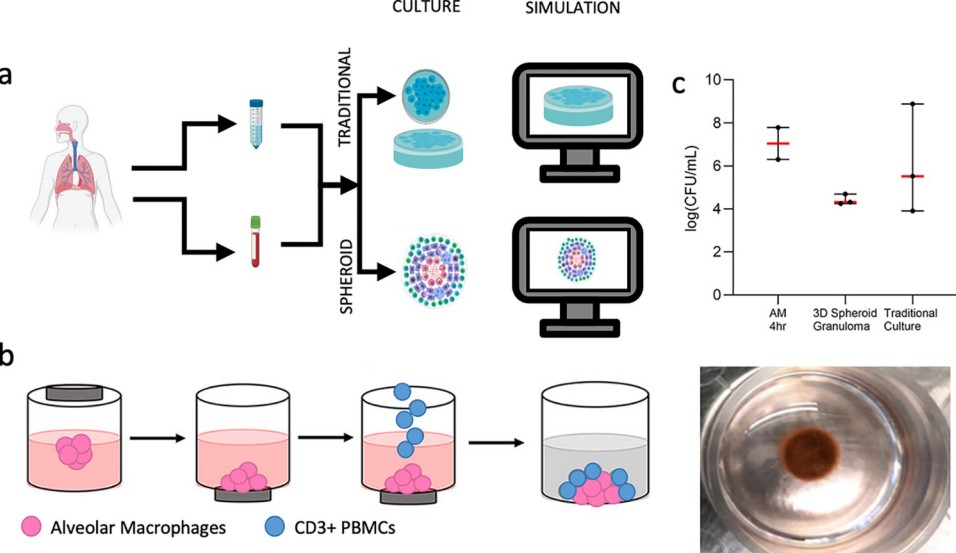

**Fig 1. Outline of *in vitro* experimental procedure.** a) Patients with active TB have bronchoalveolar fluid (BALF) and peripheral blood collected. Alveolar macrophages (AM) are isolated from BALF and PBMCs from peripheral blood. AM and PBMCs are recombined *in vitro* to create traditional and spheroid *in vitro* models, along with corresponding *in silico* traditional and spheroid simulations. b) For the 3D spheroid models, AM are infected and combined with magnetic beads before being magnetically levitated. Autologous T cells (i.e., the CD3+ fraction of the PBMCs) are added at day 2, forming a cuff around the macrophage core. This produces a spheroid of cells that is floating in the center of the well. c) Spheroid culture has lower CFU counts at day 6 than traditional culture. (Adapted from Kotze et al. 2021 [11]).

represent the 3D environment that pathogens and host immune cells would encounter *in vivo* [8–10]. The flat plastic or glass used to generate 2D *in vitro* models differs from the *in vivo* tissue microenvironment, and the architecture of the system is lost along with certain cell-cell and cell-extracellular matrix interactions [8,10]. Cell shape, surface protein distribution, migration, and even RNA transcription levels can all be influenced by the dimension of the model [8,11]. It is thought that 3D cultures will better recreate tissue characteristics, thereby improving both physiological relevance and success of drug screening *in vitro* [8–10,12].

3D cultures can be created either by using scaffolds for growing cells in 3D such as hydrogels or combinations of extracellular matrix proteins and polymers, or by making organoids/spheroids [9]. Both of these approaches have been used in the context of modeling *Mtb* infection *in vitro* [11,13,14]. Collagen-alginate and collagen-agar models with peripheral blood mononuclear cells (PBMCs) have been developed to explore the impact of collagen on *Mtb* infected cells and to create a prolonged culture system used to study the impact of immuno-modulatory therapies [13,14].

We developed an *in vitro* biomimetic 3D spheroid granuloma model [11]. Patient derived alveolar macrophages are infected with *Mycobacterium bovis* BCG, and magnetic nanospheres are used to levitate the cells and form a macrophage spheroid (Fig 1A and 1B). Autologous adaptive immune cells isolated from PBMCs are added at day 2 of the 6 day culture, and these cells self-assemble around the outside surface of the macrophage spheroid. When comparing this spheroid granuloma model to a corresponding traditional monolayer culture, we found the spheroid was better able to control bacteria (Fig 1C) [11]. In a computational model of the spheroid and traditional cultures, we found that one *in silico* model structure was able to represent both culture setups with shared parameters, that macrophage activation is dependent on IFN-γ from infiltrating T cells, and that bacterial growth could be controlled in a structure-

dependent or -independent manner [15]. Here, we aim to use this computational model to identify potential mechanistic reasons for the experimentally observed difference in bacterial control between spheroid and traditional cultures. Not only will this help us better understand how the structure of cultures impacts immune responses, but this can also provide insights into how to engineer more complex *in vitro* models.

Our established computational simulations of both the traditional and spheroid systems are used to predict the mechanisms leading to differential bacterial control. We hypothesize that spatial differences between spheroid and traditional models alone can account for changes in bacterial control. We predict that macrophage activation will be higher in the spheroid simulation to account for these differences in bacterial load. We tested this hypothesis by assessing activation levels and proximities of cells in the simulations. Further, we used virtual knockouts (KOs) (virtual elimination of individual parameters or mechanisms) to estimate the contributions of different T cell populations to bacterial control. Lastly, we uncouple the layered structure and 3-dimensional movement to predict if either is independently responsible for bacterial control. This computational analysis allows us to generate hypotheses about the mechanisms responsible for differential bacterial control, which can then be further tested *in vitro*.

## 2. Methods

### 1.1 Experimental methods

The *in vitro* cultures that are being compared are described in detail in Kotze et al. [11]. Briefly, suspected TB patients were enrolled in the study. Bronchoalveolar fluid was collected with bronchoscopies performed by qualified clinicians and nursing staff according to international guidelines, and alveolar macrophages were isolated from this fluid [16]. Peripheral blood samples were taken by venipuncture into two 9mL sodium heparinized (NaHep) vacutainers, and PBMCs were isolated using the Ficoll-Paque method described previously [11]. Total CD3+ T cells were isolated from PBMC by MACS Microbead technology, as described previously [9]. First, alveolar macrophages were infected with *Mycobacterium bovis* Bacille Calmette-Guerin (BCG) at a multiplicity of infection (MOI) of 1 for 4 hours. Antimycotic antibiotic (penicillin/streptomycin/amphotericin B) supplemented media was then used to remove extracellular bacteria for an hour, with washes performed afterward. Alveolar macrophages were then treated with biocompatible NanoShuttle (m3D Biosciences Inc., Greiner Bio-One). The spheroid culture was established by the addition of a magnetic neodymium drive to levitate the alveolar macrophages, allowing the formation of an initial spheroid. Alveolar macrophages without a levitation drive were used to establish a traditional monolayer culture (Fig 1A). After 48 hours, autologous T cells were added to both the monolayer and 3D spheroid cultures. In the spheroid culture, an organized structure developed, with T cells forming a cuff around the macrophages (Fig 1B). In traditional monolayers, macrophages and T cells randomly settled to the bottom of the culture well. Thus, both traditional and spheroid cultures were created using the same patient cells, ratio of cells, bacterial strain, MOI, and timing of T cell addition. After 6 days of culture, spheroid cultures were mechanically dissociated using gentle pipetting, to ensure single cell suspensions of both spheroid and traditional cultures. Single cells were lysed and serial dilutions were plated on Middlebrook 7H11 agar plates (BD Biosciences). Colonies were manually counted after 21 days of growth. One of the findings of this work was that spheroids were better able to control bacteria than their traditional counterpart at 6 days post infection, which is reflected in the CFU/mL measurements (Fig 1C).

## 1.2 Computational methods

The computational model established and calibrated in Petrucciani et al. [15] was used to simulate the traditional and spheroid cell cultures. The model code can be found at https://github. itap.purdue.edu/ElsjePienaarGroup/TB-in-vitro release v1.0.1. Briefly, this computational model is an agent-based model (ABM) with bacteria, macrophages, CD4+ and CD8+ T cells represented as agents. Each of these agent types has an established set of biology-based rules to govern their behavior. An overview of these rules can be found in Petrucciani et al. Fig 2. [15] Bacteria grow and divide with rates dependent on whether the bacteria are extracellular or localized within a macrophage. Macrophages, CD4+ and CD8+ T cells are all subtypes of immune cells. These immune cells move probabilistically up a chemokine gradient, secrete cytokines when activated, and age and die with set lifespans. Chemokine secretion is based on the infection and activation status of the immune cell and estimated initial ranges of these values are drawn from literature. [17] These chemokines then diffuse throughout the simulated volume and decay, which is represented with a partial differential equation model [18].

Macrophages move around the environment and phagocytose bacteria with a set probability. If a macrophage internalizes one or more bacteria it becomes an infected macrophage. Infected macrophages no longer move but can kill the intracellular bacteria. If an infected macrophage is unable to control the infection and reaches a certain limit of intracellular bacteria, it bursts, releasing the bacteria into the environment. Both macrophages and infected macrophages can be activated, which increases their phagocytosis and killing abilities. Activation is a two-step process, requiring NF-κB and STAT1 signaling. NF-κB signal can be activated in one of three ways: interacting with an activated CD4+ T cell, a local TNF-α concentration above a threshold, or a high number of local bacteria. STAT1 is activated by IFN-γ, a cytokine secreted by activated T cells.

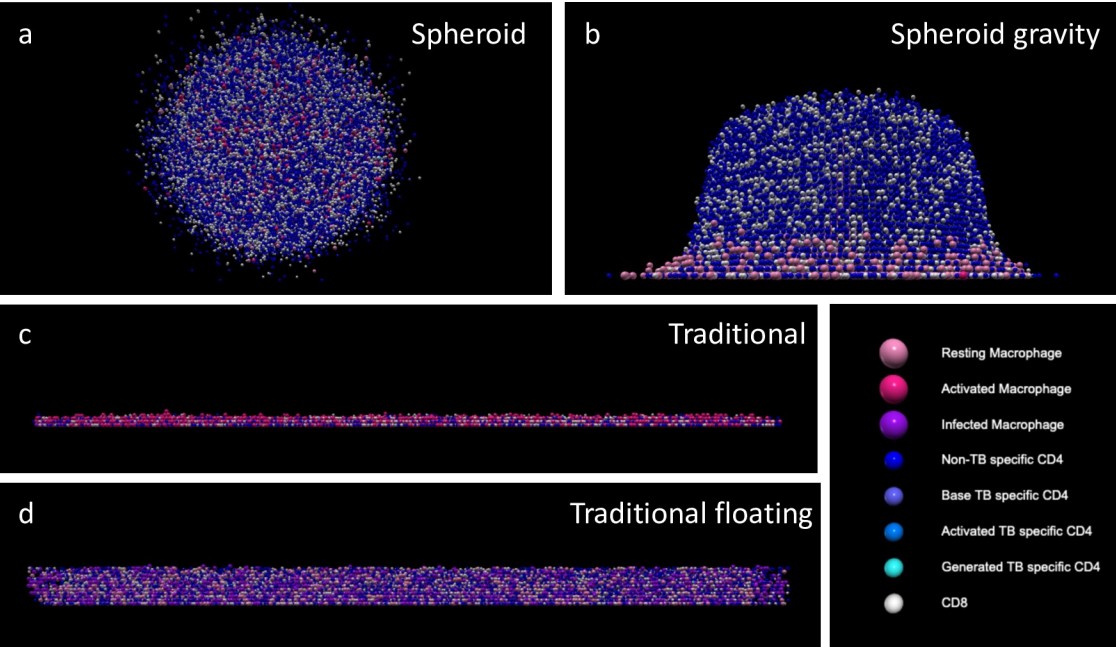

**Fig 2.** Example spatial layout of 4 simulation cases: a) spheroid structure with 3D movement, b) spheroid structure with gravity-limited movement, c) no structure with gravity-limited movement seen from the side, and d) no structure with 3D movement seen from the side. All simulations are done with 3 dimensions, but the ones with gravity-limited movement represent experimental "2D" models.

T cells are subdivided into CD4+ and CD8+ T cells. Both of these cell types follow immune cell rules and can be divided into TB-specific and non-TB-specific classes. TB-specific T cells can be activated and will proliferate when activated. CD4+ T cells are activated when they interact with a macrophage that has interacted with a bacterium. This is the equivalent of antigen presentation on MHC II. When CD4+ T cells are activated, they can activate NF-κB for macrophages through direct interaction or STAT1 though IFN-γ secretion. CD8+ T cells can be activated when they interact with a macrophage that has interacted with a bacterium and is also STAT1 activated. When CD8+ T cells are activated, they secrete IFN-γ like CD4+ T cells and can kill infected macrophages through their cytotoxic functionalities.

The difference in the two *in vitro* models of interest is simulated by adjusting the movement rules and the initial structure in our simulation. The traditional simulation has "gravity-limited" movement rules. Cells in this environment cannot float and can only move up the z-direction if they are moving on top of another cell. Neighboring cells are checked to ensure that they will meet these criteria. If the criteria are not met, then the grid square is removed from potential locations to move. All unoccupied and valid locations are then used to probabilistically move up the chemokine gradient if the concentration threshold for directed movement is met. In contrast, the spheroid simulation cells are not limited by gravity and can move in three dimensions. This means that all unoccupied grid squares are valid for probabilistic chemotaxis and random movement.

The initial structure of the cells in traditional and spheroid simulations also differs. The spheroid simulation is initiated as a sphere of macrophages in the center of the grid and the movement rules allow cells to move in any direction randomly or in response to chemokine gradients. In contrast, the traditional simulation has macrophages spread out throughout the grid space initially, with the movement rules resulting in the cells 'sinking' to the bottom of the grid (representing the bottom of the cell culture plate). When T cells are added to the spheroid simulation, they are placed in a spherical shell around the macrophage core. In the traditional simulation, T cells are distributed throughout the simulation space and, as with macrophages, allowed to 'sink' to the bottom.

## 1.3 Calibration and parameter selection

This hybrid model was parameterized in Petrucciani et al. with experimental data for bacterial fold change, cell count, and cell viability at day 6 from both the spheroid and the traditional cultures [15]. Calibration was done by performing a large initial sweep of the parameter space with 1000 Latin hypercube samples and 7 replicates. High traditional bacterial fold change only occurred in a few simulations, so iterative enrichment of these regions was performed. This was done by selecting enrichment points, narrowing the parameter space around these points, and repeating this sampling and enrichment process until increases in the desired output (i.e., traditional bacterial fold change) no longer improved. Please refer to Petrucciani et al. 2024 for more details and an uncertainty analysis. [15] The final collection of calibrated parameter sets are then composed of all simulations from all iterations that produced outputs within the experimental ranges for our 6 criteria (spheroid and traditional bacterial fold change, cell counts, and cell viability). A full list of parameters, units, and initial ranges can be found in Table 1. Parameters in the text will be italicized for clarity. Distributions of parameters within various groups of simulations can be seen in S1 Fig.

## 1.4 Virtual knockout simulations

Virtual elimination of individual parameters or mechanisms (i.e., virtual knockout) simulations were performed by setting individual model parameters to zero. Whenever virtual

**Table 1. Parameters that are varied during calibration.** Initial ranges are either determined by literature, estimated through preliminary simulations (e), or broadened to the full mathematically possible range (f). The collection of calibrated parameter sets can be found in the supplementary material: parameter descriptions are given in S1 Table. This table is reproduced from Petrucciani et al. [15].

| Parameter | Initial Range | Units | Refs |
|---|---|---|---|
| **Bacteria** | | | |
| *mtbInternalDoublingTime* | 23,69 | Hours | [19] |
| *mtbExternalDoublingTime* | 23,69 | Hours | [19] |
| **Macrophages** | | | |
| *activatedMacrophageProportion* | 0,0.1 | Activated macrophages/ Total macrophages | e |
| *baseKillingProbability* | 0.0001,0.02 | Per tick | e |
| *activeKillingProbability* | 0.002,0.3 | Per tick | e |
| *basePhagocytosisProbability* | 0,1 | Per tick | f |
| *activePhagocytosisProbability* | 0,1 | Per tick | f |
| *phagocytosisThreshold* | 8,12 | Internal bacteria | [20] |
| *cellularDysfunctionThreshold* | 8,12 | Internal bacteria | [20] |
| *nfkbSpan* | 0.16,166 | Hours | [18] |
| *TNFthresholdForNFkBActivation* | 40,500 | Molecules | e |
| *bacThresholdForNFkBActivation* | 20,150 | External bacteria | [20] |
| *stat1Span* | 0.16,166 | Hours | [18] |
| *IFNthresholdForStat1Activation* | 40,500 | Molecules | e |
| *ActivatedMacrophageTNFSecretion* | 0,40 | Molecules/second | [17] |
| *InfectedMacrophageTNFSecretion* | 0,40 | Molecules/second | [17] |
| *macrophagePopulation_MaxLifespan* | 20,100 | Days | [20] |
| *macrophagePopulation_MaxActivatedLifespan* | 7,13 | Days | [20] |
| *baseMovementProbabilityMacro* | 0.5,1 | Per tick | [21–24] |
| *activatedMovementProbabilityMacro* | 0,0.5 | Per tick | e |
| **CD4+ T cells** | | | |
| *fractionCD4* | 0.5,0.65 | CD4+ T cells/ CD3+ T cells | [25,26] |
| *fractionTBSpecific* | 0.0001,0.06 | TB specific CD4+ T cells/ Total CD4+ T cells | [27,28] |
| *activatedTBSpecificCD4Fraction* | 0,0.1 | Initial activated TB specific CD4 T cells/ Total TB specific CD4 T cells | e |
| *CD4ActivationProbability* | 0,1 | Per tick | f |
| *CD4DeactivationProbability* | 0,1 | Per tick | f |
| *ActivatedCD4TNFSecretion* | 0,40 | Molecules/second | [17] |
| *ActivatedCD4IFNSecretion* | 0,40 | Molecules/second | [17] |
| *cd4PopulationDoublingTime* | 6,16 | Hours | [29,30] |
| *maximumCD4Generations* | 3,10 | Generations | [29,31,32] |
| *cd4Population_MaxLifespan* *cd8Population_MaxLifespan* | 34,340 | Days | [33–35] |
| *cd4Population_ActivatedLifespan* *cd8Population_MaxActivatedLifespan* | 2.5,4 | Days | [20,29] |
| *baseMovementProbabilityCD4* *baseMovementProbabilityCD8* | 0,1 | Per tick | f |
| *activatedMovementProbabilityCD4* *activatedMovementProbabilityCD8* | 0,1 | Per tick | f |
| **CD8+ T cells** | | | |
| *CD8Fraction* | 0.3,0.35 | CD8+ T cells/ CD3+ T cells | [26] |
| *tbSpecificCD8Fraction* | 0.0001,0.06 | TB specific CD8+ T cells/ Total CD8+ T cells | [27,28] |
| *activatedTBSpecificCD8Fraction* | 0,0.1 | Initial activated TB specific CD8 T cells/ Total TB specific CD8 T cells | e |
| *CD8ActivationProbability* | 0,1 | Per tick | f |
| *CD8DeactivationProbability* | 0,1 | Per tick | f |
| *ActivatedCD8TNFSecretion* | 0,40 | Molecules/second | [17] |

*(Continued)*

**Table 1.** (Continued)

| Parameter | Initial Range | Units | Refs |
|---|---|---|---|
| *ActivatedCD8IFNSecretion* | 0,40 | Molecules/second | [17] |
| *cd8PopulationDoublingTime* | 3,13 | Hours | [30] |
| *maximumCD8Generations* | 7,20 | Generations | [31,32,36] |
| *CD8KillProbability* | 0.012,0.12 | Per tick | [20] |
| **Diffusion** | | | |
| *TNFthresholdForImmuneCellMovement* | 1,500 | Molecules | e |
| *TNFDiffusionCoefficient* | 0.1,1 | 10^-7 cm^2/s | [18] |
| *TNFDegradationRatePerSecond* | 0.96,10 | 1/s | e |
| *IFNDiffusionCoefficient* | 0.1,1 | 10^-7 cm^2/s | [18] |
| *IFNDegradationRatePerSecond* | 0.96,10 | 1/s | e |
| *granulomaFractionOfDiffusion* | 0,1 | - | f |
| *sphereEfficiency* | 0.65,0.9 | - | e |

knockouts are mentioned, it refers to simulations. Four different knockout scenarios were simulated. First, IFN-γ secretion from CD4+ T cells was eliminated. This was accomplished by running the simulation with an *ActivatedCD4IFNSecretion* parameter value of zero. Next, IFN-γ secretion from CD8+ T cells was knocked out. This was similarly accomplished by setting *ActvatedCD8IFNSecretion* to zero. These two knockouts were also simulated in tandem. This double knockout lacked any IFN-γ, meaning there was no macrophage activation. Finally, CD8+ cytotoxic killing was eliminated by setting *CD8KillProbability* to zero. These knockouts were compared with each other and the default simulations to quantify the relative contributions of these mechanisms to bacterial control and macrophage activation.

## 1.5 Movement rule swap

To assess the impact of movement restrictions or initial structure on infection progression, we computationally uncouple these two mechanisms. The default simulations are for spheroid and traditional setups. The spheroid simulation has 3D movement rules and the imposed spheroid structure (Fig 2A), while the traditional simulation has gravity-limited movement rules and no structure (Fig 2C). We simulate 2 new scenarios: one with 3D movement rules and no structure (traditional floating, Fig 2D) and one with gravity-limited movement and imposed structure (spheroid gravity, Fig 2B). These new simulation setups are compared with the two default setups to quantify the impact of movement and structure on bacterial control, macrophage activation, and T cell activation.

## 2. Results

### 2.1 Calibrated simulations predict heterogeneous bacterial dynamics that are consistent with experimental observations

In Petrucciani et al., both spheroid and traditional simulations were initialized to reflect the corresponding experimental conditions as described in Section 2.1 and were calibrated to experimental data at day 6 [15]. A brief description of this calibration process is given in Section 2.3, but full details can be found in the previous work. Parameters are assumed to be the same between traditional and spheroid simulations. Here, calibrated simulations are examined to identify the causative mechanisms of bacterial control in spheroid and traditional simulations. Heterogeneity is observed in bacterial and host immune dynamics in these calibrated simulations because there is 1) a lot of uncertainty between initial and end time points where

experimental data are not available and 2) the experimental data at the end time point has a large range/spread. Three qualitatively different behaviors emerge from this collection of simulations.

To separate the qualitatively different behaviors, a point of separation is selected visually (Fig 3A) to separate the dynamic behaviors of bacterial counts: parameter sets where bacterial loads follow similar trajectories between spheroid and traditional simulations (similarly controlled simulations), vs. parameter sets where bacterial load diverges between spheroid and traditional simulations (differentially controlled simulations). Due to the overlap in the experimental ranges of CFU (Fig 3A) between spheroid and traditional data, some of our simulations have similar bacterial fold changes for the traditional and spheroid simulations (Fig 3B). As expected, aggregate cellular dynamics (such as activated macrophages and activated T cells) are also similar for the spheroid and traditional similarly controlled simulations (S2 Fig). Other simulations show that it is possible to have differential control of bacterial growth between the traditional and spheroid conditions (Fig 3D and 3E). The spheroid and traditional CFU counts in these differentially controlled simulations begin diverging around days 3 and 4, and this divergence is only due to differences in initial structure and cell movement (in 3D vs. restricted to the bottom of the culture plate, respectively).

These groups (similarly vs differentially controlled) correspond closely to separating the simulations into groups with low and high probability for an internalized bacteria to be killed by a resting macrophage (*baseKillingProbability*) (Fig 3C). Simulations where bacteria were similarly controlled in traditional and spheroid conditions have higher *baseKillingProbability*. This implies that in these similarly controlled simulations, resting macrophages are contributing more to controlling bacterial growth, and therefore these simulations are expected to depend less on adaptive immune parameters (such as the fraction of TB specific CD4s). The differentially controlled simulations have lower *baseKillingProbability*, meaning decreased killing ability of resting macrophages. This suggests that bacterial control is more dependent on the adaptive immune system following the addition of T cells at day 2; and that the added T cells are more effective in the spheroid simulations than the traditional simulations.

The differentially controlled simulations are further subdivided into two groups. Several parameters show bimodal distributions in their estimated values for differentially controlled simulations. The bimodal distributions correspond across these parameters (i.e., for all the simulations that have high parameter 1, they all have low parameter 2 and vice versa) (S1 Fig). Thus, the differentially controlled simulations are split into a group with *cd4doublingtime* less than 11 hours (differentially controlled group 1, DC-1) and *cd4doublingtime* greater than or equal to 11 hours (differentially controlled group 2, DC-2) as this is between the two peaks (S1 Fig). These same two groups are also found if dividing between the two peaks based on *maximumCD8Generations* at 14.8, *phagocytosisThreshold* at 9.1, *ActivatedCD4IFNSecretion* at 26, or *ActivatedCD4TNFSecretion* at 32. When these groups are separated based on *cd4doublingtime*, they also exhibit different dynamic behaviors. DC-1 simulations have two major decreases in bacterial count on day 3 and day 5 (Fig 3D), while DC-2 simulations have one smoother decrease (Fig 3E).

DC-1 parameter trends include impaired CD8+ T cell behaviors (lower maximum CD8+ T cell generations, lower activated TB-specific CD8+ T cell fraction, lower TB-specific CD8+ T cell fraction, higher CD8+ T cell deactivation probability), improved CD4+ T cell behaviors (higher CD4+ T cell activated lifespan, higher activated TB-specific CD4+ T cell fraction, higher CD4+ T cell IFN and TNF secretion), and some parameters that don't fit either of these groupings (lower STAT1 span, lower macrophage maximum life span, higher granuloma fraction of diffusion, higher CD8+ kill probability, slightly lower bacterial external doubling time, lower phagocytosis threshold, higher lifespan for activated macrophages). The distributions of

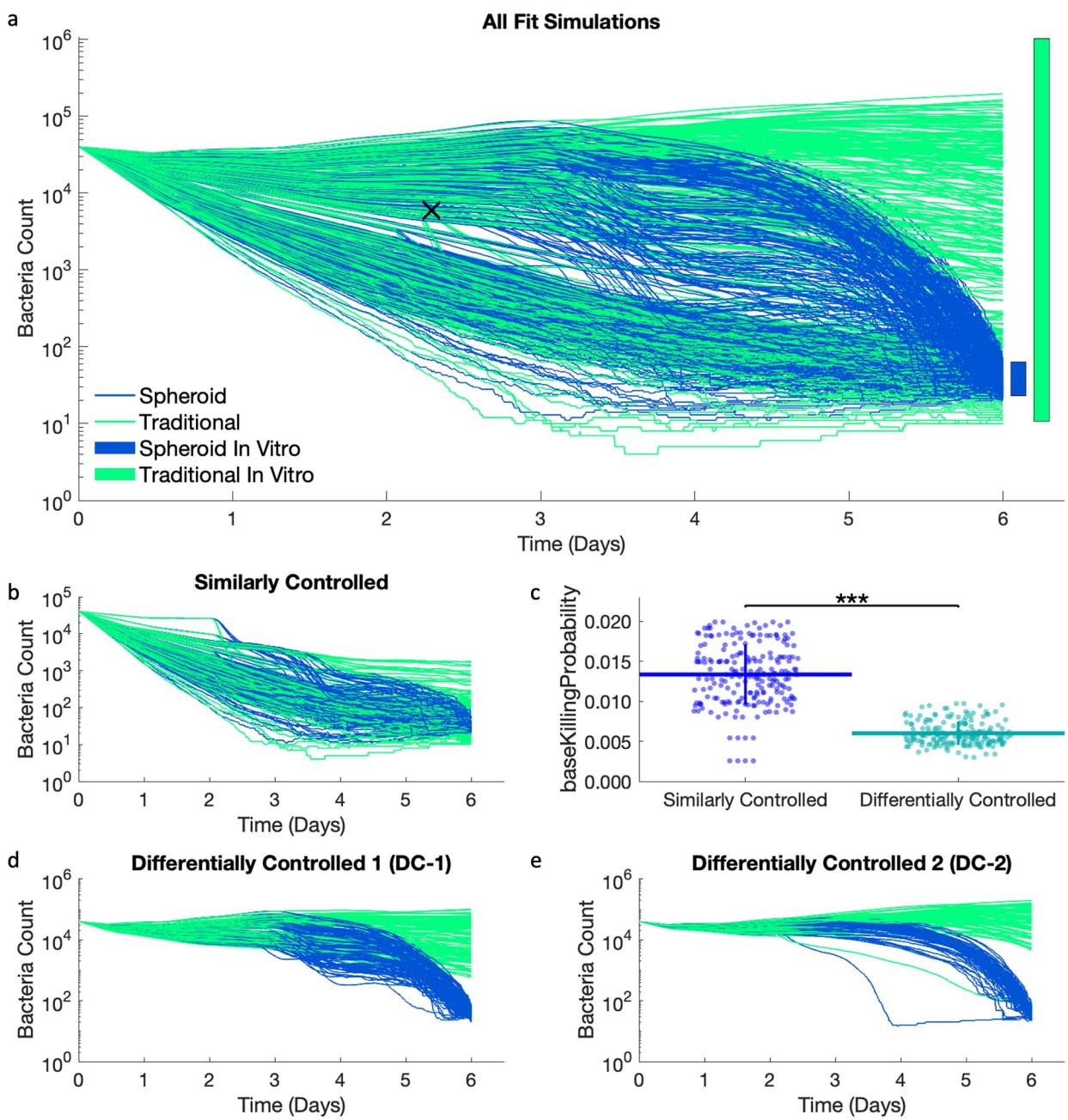

**Fig 3.** Spheroid (blue) and traditional (green) simulations have been calibrated in a paired manner to in vitro data. a) Bacterial count dynamics for spheroid and traditional simulations over the 6-day time course. The blue and green bars on the right-hand side represent the ranges of experimental calibration data (calculated from fold changes). The black x is the point used to differentiate between similarly and differentially controlled simulations. We divide the simulations into two groups with traditional CFU greater than 6000 (differentially controlled simulations) or less than or equal to 6000 (similarly controlled simulations) at time step 550 (about day 2.29). b) Due to the overlap in the experimental data for CFU, some spheroid and traditional simulations perform similarly. c) The similarly controlled runs have a higher baseKillingProbability than the differentially controlled runs. Two patterns emerge in the differentially controlled simulations: d) simulations with two major decreases and e) simulations with one major decrease.

the parameters for these two groups can be seen in S1 Fig. Many of these parameters show inverted relationships between CD4+ and CD8+ T cell behaviors, suggesting that the distinction between these groups is which T cell fraction is dominant in count or behavior. These

could represent natural variation between individual or different immune statuses. For example, DC-1 could represent people with exhausted CD8+ T cells due to chronic inflammation, and DC-2 could represent people with reduced CD4+ T cell counts due to HIV infection or iatrogenic immunosuppression.

Overall, our results illustrate that many different bacterial dynamic behaviors can be consistent with the experimental data. These diverse bacterial dynamics are driven by diverse parameter ranges, which could represent either a) different host immune characteristics or b) large degrees of freedom in our simulation parameters due to limited time resolution in the experimental data. In the following sections, these two groups (DC-1 simulations and DC-2 simulations) are explored further, through the comparison of spheroid and traditional setups, to predict possible mechanistic reasons for improved bacterial control in the spheroid simulation compared to the traditional simulations.

## 2.2 The divergence of bacterial load in spheroid and traditional DC-1 simulations is due to increased proportions of activated macrophage killing in spheroid simulations

DC-1 simulations show an initial divergence in bacterial load between day 3 and day 4 (Fig 4A). Although the bacterial counts continue to diverge from day 4 to 6, we examine simulation outputs at the time of this initial divergence to identify the driving mechanisms of the initial divergence.

Bacteria in the simulation can be killed in 3 different ways: killed by resting macrophages, killed by activated macrophages, and killed along with infected macrophages by CD8+ T cells. The proportions of total bacterial killing due to each of these death methods are compared between spheroid and traditional simulations (Fig 4B-4D). During our time period of interest (day 3 to 4), spheroid simulations have a significantly higher proportion of bacterial killing by activated macrophages. This corresponds to an equivalent lower proportion of bacterial killing by resting macrophages in the spheroid simulations compared to traditional simulations. No significant differences in cytotoxic CD8+ T cell killing are seen between the two simulation setups, with minimal cytotoxic killing observed in both spheroid and traditional simulations between day 3 and 4.

A higher proportion of activated macrophage killing in the spheroid simulations prompted a closer look at macrophage activation at day 4. The total number of activated macrophages is significantly higher in the spheroid than in the traditional simulation (Fig 4E). This difference in macrophage activation is due to STAT1 rather than NF-κB, as the spheroid has significantly more STAT1 activated macrophages and significantly fewer NF-κB activated macrophages than the traditional simulation (Fig 4F and 4G). This is constant with the previously-performed uncertainty analysis that showed a correlation between higher IFN-γ secretion and lower bacterial counts [15]. The fact that the spheroid has more STAT1 activated macrophages implies that the spheroid culture should have more IFN-γ, which activates STAT1, and more activated T cells, which produce IFN-γ. However, fewer activated T cells are seen in the spheroid simulation (Fig 4H–4J). When these activated T cells are subdivided into CD4+ and CD8+ T cells, less activation is still seen in both CD4+ and CD8+ T cell populations in the spheroid simulations. These trends in macrophage and T cell activation are consistent from day 4 to 6, with the spheroid simulations having more activated macrophages even with fewer activated T cells (S3 Fig). These results suggest that the improved bacterial control in the spheroid simulation is due to increased macrophage activation, specifically STAT1 activation, that cannot be accounted for by bulk differences in T cell activation.

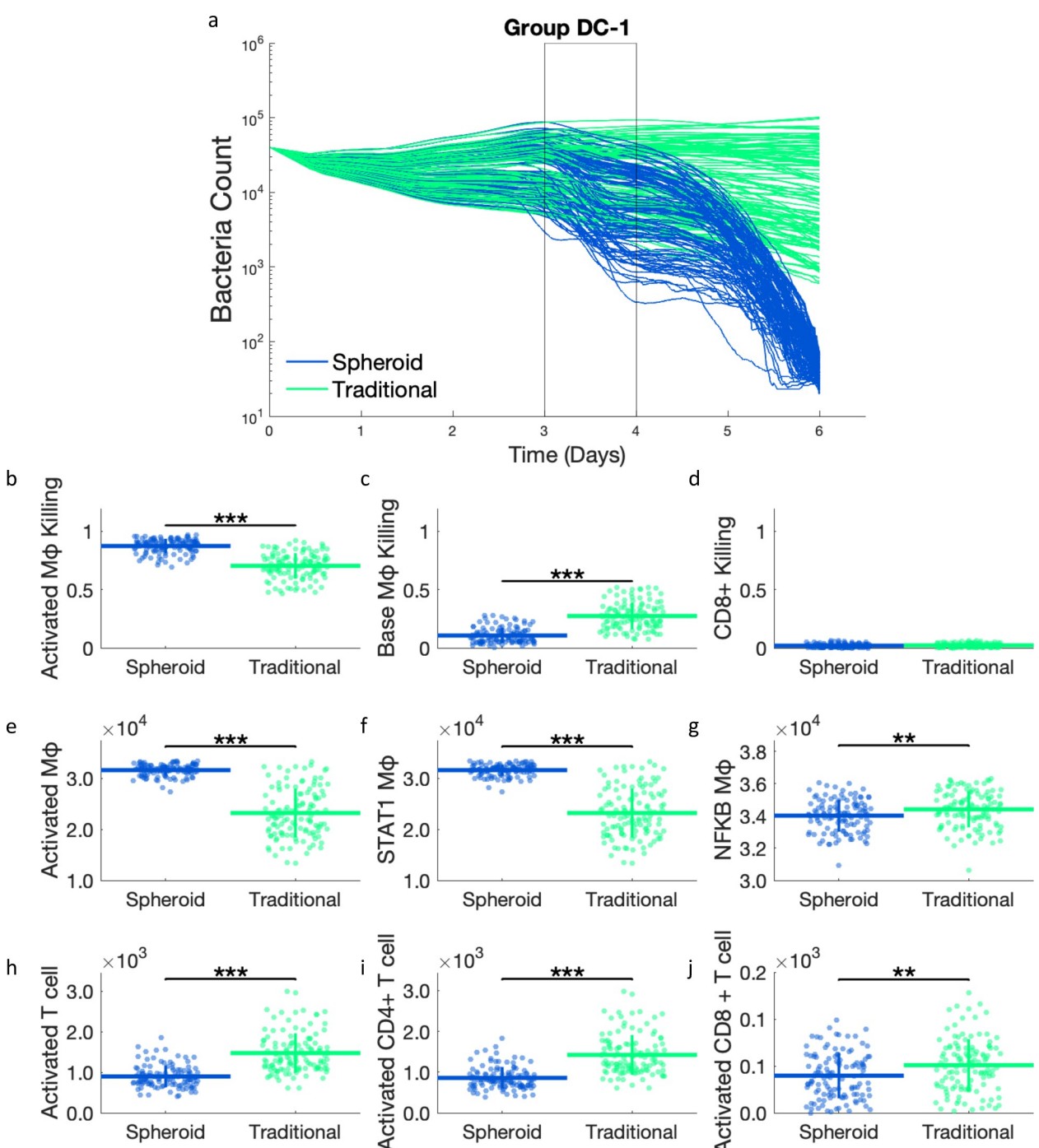

**Fig 4. a) Bacterial dynamics for DC-1 simulations.** The time range of interest (day 3–4) is outlined. These are the same time courses as shown in Fig 3C. Proportions of bacterial killing due to b) activated macrophages, c) base macrophages, and d) CD8+ T cell killing from day 3 to 4. Comparing total counts of e) activated macrophages, f) STAT1 activated macrophages, g) NF-κB activated macrophages, h) activated T cells, i) activated CD4+ T cells, and j) activated CD8+ T cells between the spheroid and traditional simulations at day 4. ** p< = 1e-2, *** p< = 1e-3.

### 2.3 In DC-1 simulations, fewer activated T cells are able to activate more macrophages due to spatial distribution in the spheroid simulations

In the previous section, we note a contradiction between day 3 and 4 when bacterial loads in the spheroid and traditional simulations start to diverge, where fewer activated T cells are able to activate more macrophages in the spheroid simulations. Because the difference in STAT1 activation of macrophages is not due to bulk numbers of T cells, we next investigate the simulated spatial layout of macrophages and T cells. To determine how close macrophages are to the nearest T cells, the closest T cell neighbor of each macrophage is found. A K-D tree is built using the locations of T cells [37]. This K-D tree is then queried using each macrophage location to find the nearest T cell. These distances are averaged over all macrophages in each simulation. The distribution of these means is compared for spheroid and traditional simulations, representing the average distance from macrophage to T cell (Fig 5). This process is repeated, constructing K-D trees with different subsets of T cells (i.e., activated, activated CD4+, activated CD8+) to find the average distances between macrophages and these subsets. Macrophages in spheroid simulations are farther away from T cells in general (Fig 5A), but are closer to IFN-γ producing activated T cells (Fig 5B). This suggests that although there are fewer activated T cells in the spheroid simulations, the macrophages are more closely colocalized to IFN-γ producing T cells leading to more macrophage activation. This trend holds true for both activated CD4+ and CD8+ T cells (Fig 5C and 5D), suggesting they both contribute to macrophage activation. These trends are consistent across days 3 to 6 (S4 Fig). Thus, our results indicate that increased colocalization of macrophages with activated CD4+ and CD8 + T cells in the spheroid simulations is associated with more macrophage activation and more bacterial killing by activated macrophages. This colocalization can be visualized in a representative spheroid simulation at day 4 (Fig 5E and 5F). The activated T cells are primarily located at the edge of the macrophage core. This proximity leads to more macrophage activation and more bacteria killing. Our simulations therefore suggest that for DC-1 simulations, the spatial layout in spheroids allows for better bacterial control.

### 2.4 In DC-2 simulations, spheroid simulations have higher proportions of bacterial death due to higher CD8+ T cell activation, in addition to activated macrophage killing

DC-2 simulations do not have same the initial decrease in bacterial load at day 3 that we observe in DC-1 simulations. Instead, DC-2 simulations have a shallower slope downwards starting at day 3 and continuing for the rest of the simulation (Fig 6A). As the spheroid and traditional simulation initially start to diverge at day 3, the proportions of bacterial killing from day 3 to 4 are analyzed (Fig 6B-6D). Similar to DC-1 simulations, more bacterial killing is found to be due to activated macrophages and less killing due to base macrophages in the spheroid simulation compared to traditional simulations. However, in contrast to DC-1 simulations, DC-2 simulations also show an increased proportion of killing due to CD8+ T cell killing. DC-2 parameter trends include higher TB-specific CD8+ T cell fraction and CD8+ T cell deactivation probability, which are both correlated with increased bacteria killed by CD8+ T cells in our previous uncertainty analysis [15]. As in DC-1 simulations, more activated macrophages are seen in the spheroid simulation with STAT1 as the limiting factor. However, in this case, we do see more activated T cells in spheroid simulations (Fig 6H). When subdivided into CD4+ and CD8+ T cells, we see that higher T cell activation in spheroids is primarily due to higher CD8+ T cell activation (Fig 6I and 6J). Thus, in DC-2 simulations, the bulk difference in T cell activation can account for the increase in macrophage activation and activated macrophage killing, and the bulk difference in CD8+ T cell activation can account for the increase in

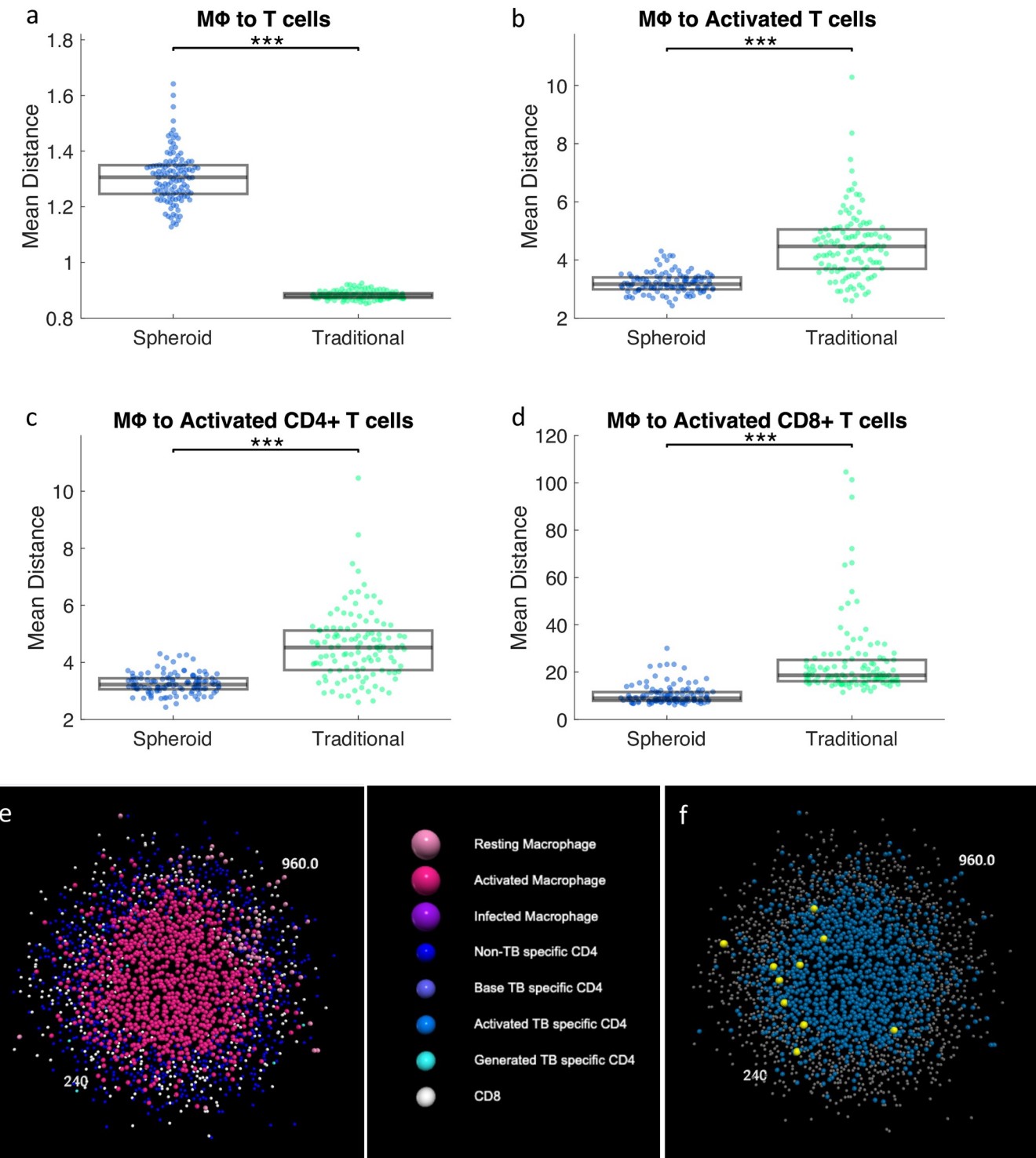

**Fig 5.** The mean distance from macrophages to nearest a) T cells, b) activated T cells, c) activated CD4+ T cells, and activated CD8+ T cells. Each dot represents an average of all cells in a single simulation. The distribution of the mean distances across the set of simulations is compared between spheroid and traditional at day 4. e) A slice through the center of a representative spheroid simulation from the DC-1 simulations at day 4 colored by cell type. f) The same simulation with macrophages in blue, activated T cells enlarged and in yellow, and all other cells in grey. ** p< = 1e-2, **** p< = 1e-4.

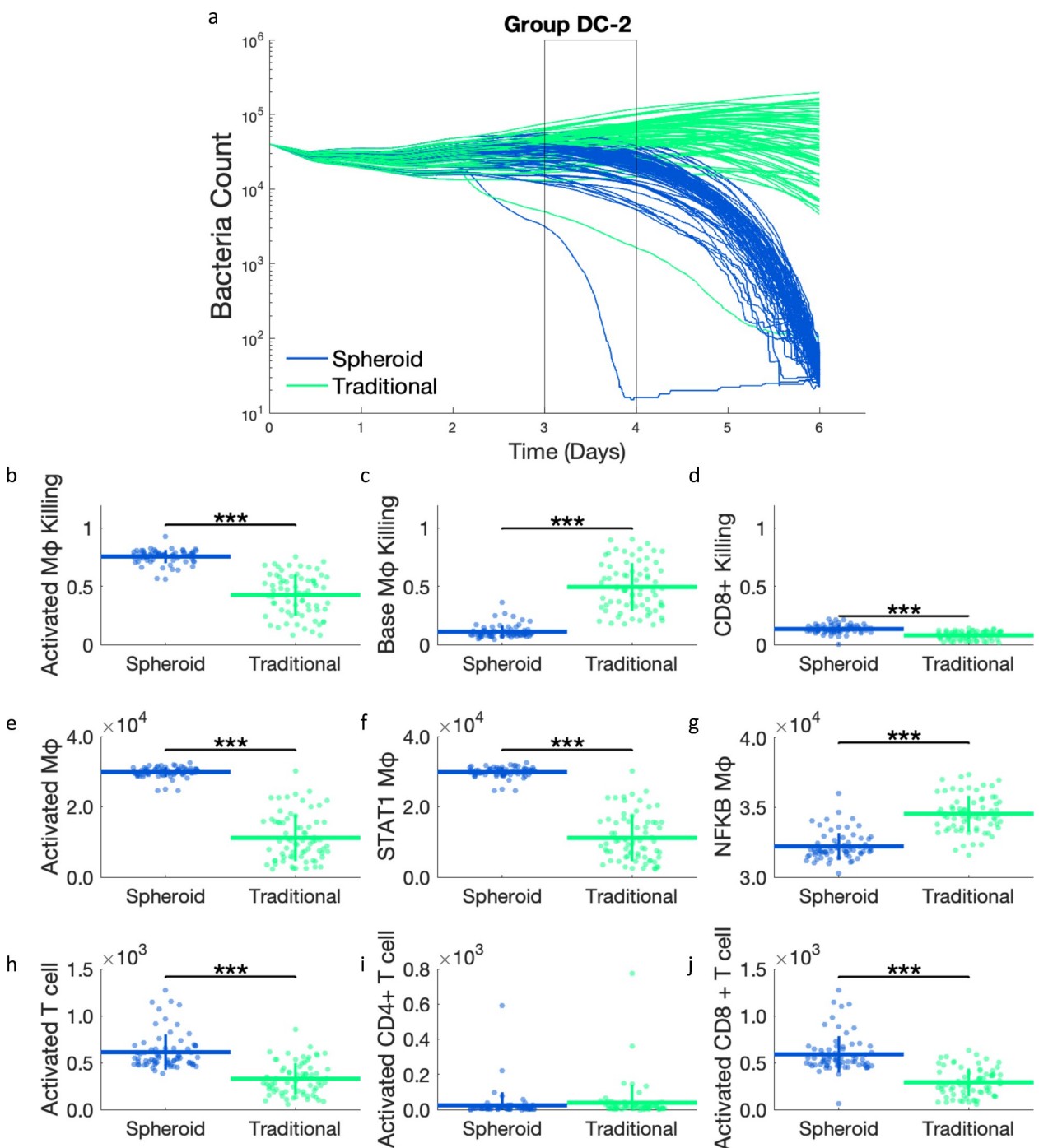

**Fig 6.** a) Bacterial count over time for DC-2 simulations. The initial deviation of spheroid and traditional simulations occurs during day 3 to 4 (black rectangle). These are the same time courses as shown in Fig 3D. Proportions of bacterial killing due to b) activated macrophages, c) base macrophages, and d) CD8+ T cell killing from day 3 to 4. Comparing total counts of e) activated macrophages, f) STAT1 activated macrophages, g) NF-κB activated macrophages, h) activated T cells, i) activated CD4+ T cells, and j) activated CD8+ T cells between the spheroid and traditional simulations at day 4. *** p< = 1e-3.

CD8+ T cell killing. However, it remains unclear why more CD8+ T cells are activated in the spheroid simulations than in the traditional DC-2 simulations.

## 2.5 Increased activated CD8+ T cells in DC-2 spheroid simulations is due to closer proximity to activating macrophages

The higher proportion of bacteria killed by CD8+ T cells in DC-2 spheroid simulations corresponds to higher counts of activated CD8+ T cells (Fig 6). TB-specific (TBS) CD8+ T cells become activated when they directly interact with a macrophage that has interacted with bacteria and that is also STAT1 activated. As stated previously, STAT1 activation is higher in the spheroid simulation from day 3 onward. Therefore, one would expect that more STAT1 activated macrophages would be able to activate more TB-specific CD8+ T cells in the spheroid simulations compared to the traditional simulations. The relative trend of macrophages that have interacted with bacteria is not consistent over time though; spheroid simulations have fewer on day 2, more on day 3, and fewer again by day 5 (S5 Fig). As before, to assess the likelihood of TB-specific CD8+ T cells coming into contact with *Mtb*-interacted macrophages, we explored the spatial layout of the cells. To determine how close TB-specific CD8+ T cells are to the nearest macrophage, the closest macrophage neighbor of each TB-specific CD8+ T cell is found. A K-D tree is constructed using all of the macrophage or STAT1 activated macrophage locations [37]. This tree is used to find the closest macrophage to every TB-specific CD8+ T cell. These distances are averaged over all TB-specific CD8+ T cells in each simulation. The distribution of these means is compared for spheroid and traditional simulations (Fig 7). While TB-specific CD8+ T cells are on average further from macrophages in the spheroid simulations (Fig 7A), they are closer to the STAT1 activated macrophages (Fig 7B). This proximity can be visualized in a representative spheroid simulation at day 4 (Fig 7C and 7D). The TB-specific CD8+ T cells are localized in the cuff around the edges of the spheroid, and the STAT1 macrophages are along the edge of the macrophage core. This proximity leads to more TB-specific CD8+ activation and more cytotoxic killing. The closer colocalization of TB-specific CD8 + T cells to STAT1 activated macrophages in spheroids is consistent across the time course of the simulation (S6 Fig). These results together suggest that higher levels of STAT1 activation along with the closer colocalization of STAT1 activated macrophages with TB-specific CD8+ T cells lead to higher CD8+ T cell activation and more cytotoxic killing in the DC-2 spheroid simulations.

## 2.6 DC-1 simulations have a CD4+ T cell dominant response, while DC-2 simulations have a mixed/CD8+ T cell dominant response

In DC-1 simulations, CD4+ and CD8+ T cells both have lower bulk levels of activation, but closer proximity to macrophages. This could suggest that both subsets of T cells play a role in macrophage activation. To further dissect the role that IFN-γ derived from CD4+ T cells and CD8+ T cells plays in macrophage activation, we perform virtual knockouts (KOs) of CD4 + and CD8+ IFN-γ secretion independently and together as control. Macrophage activation is not possible in the double KO, leading to higher bacterial loads and confirming the role of macrophage activation in section 3.2. CD4+ T cell activation should be unaffected as it is upstream of the mechanism being knocked out. All four KOs have similar levels of CD4+ activation to the original simulations (S7 Fig). Macrophage activation is downstream and directly impacted by the amount of IFN-γ secreted. CD8+ T cell activation is indirectly downstream of IFN-γ secretion, as STAT1 activated macrophages activate CD8+ T cells. In DC-1 simulations, the double KO performs similarly to the KO of IFN-γ secretion from CD4+ T cells in terms of bacterial control (Fig 8A). Both of these cases have less macrophage and CD8+ T cell activation

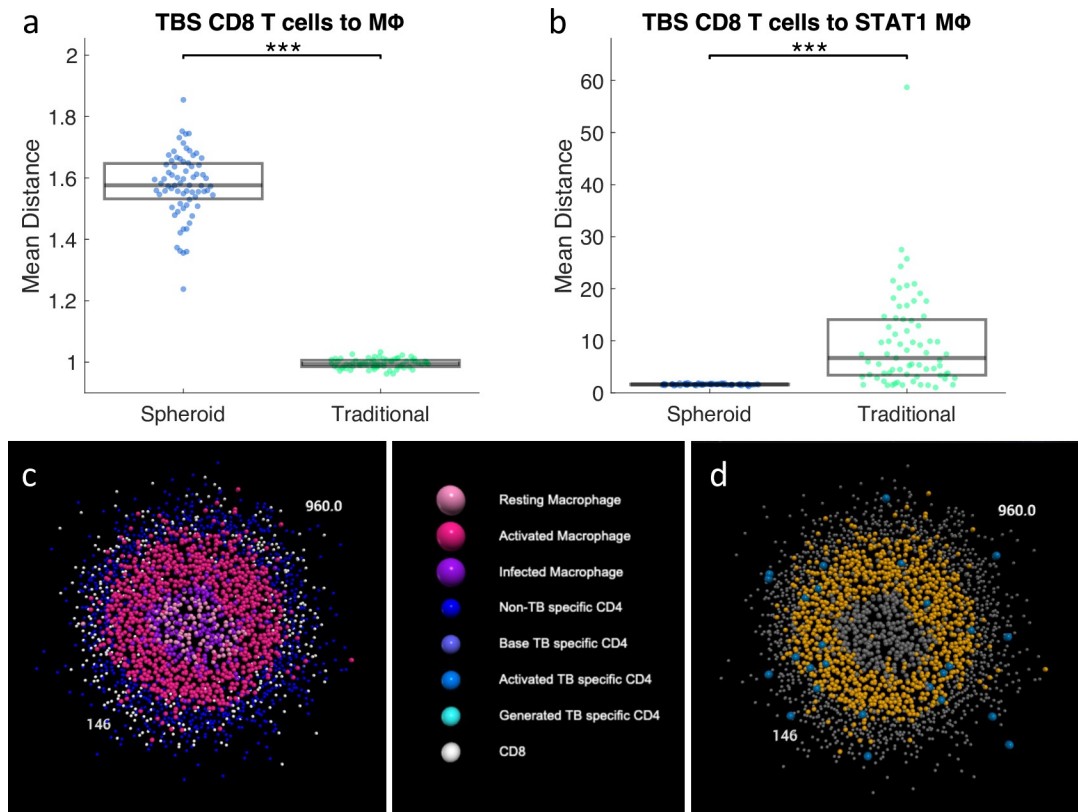

**Fig 7.** Mean distance between TBS CD8+ T cells and (a) macrophages or (b) STAT1 activated macrophages. Each dot represents the mean distance of all cells for a single simulation. c) A slice through the center of a representative spheroid simulation from the DC-2 simulations at day 4 colored by cell type. d) The same simulation with CD8+ T cells in blue, STAT1 activated macrophages in orange, and all other cells in grey. * p< = 0.05, **** p< = 1e-4.

(Fig 8D and 8G). This is in contrast to the CD8+ IFN-γ KO which has similar levels of macrophage and CD8+ T cell activation to the original scenario, despite having worse bacterial control. Taken together this suggests CD4+ T cells play a larger role than CD8+ T cells in activating both macrophages and CD8+ T cells, at least in DC-1 simulations. These results are consistent for the spheroid and traditional simulations (S7 Fig). Note that fold-changes of virtual knockout outputs relative to control outputs are not the same in spheroid and traditional simulations as there is an upper limit of bacterial growth. However, these relationships between virtual knockout outputs and control outputs trend in the same direction.

Similar KOs were performed for DC-2 simulations for completeness' sake, with the expectation that CD8+ T cells would be more important based on bulk cell counts. DC-2 simulations show similar bacterial counts between the double KO and the CD8+ T cell IFN-γ secretion KO (Fig 8B). Again, these two cases have significantly less macrophage and CD8+ T cell activation than the default scenario (Fig 8E and 8H). Thus, IFN-γ secreted from CD8+ T cells is the major activator of macrophages and, indirectly, CD8+ T cells in DC-2 simulations. This suggests some autoactivation/feedback loop of CD8+ T cells, where the IFN-γ they secrete overall leads to more CD8+ T cell activation. Local IFN-γ, both autocrine and paracrine, has been shown to enhance CD8+ T cell cytotoxic activity, motility, and survival in the context of viral infection and graft rejection [38,39].

In the DC-2 traditional simulations, CD4+ and CD8+ IFN-γ KO both have higher counts of bacteria (Fig 8C), suggesting that CD4+ T cells play a more important role in the traditional

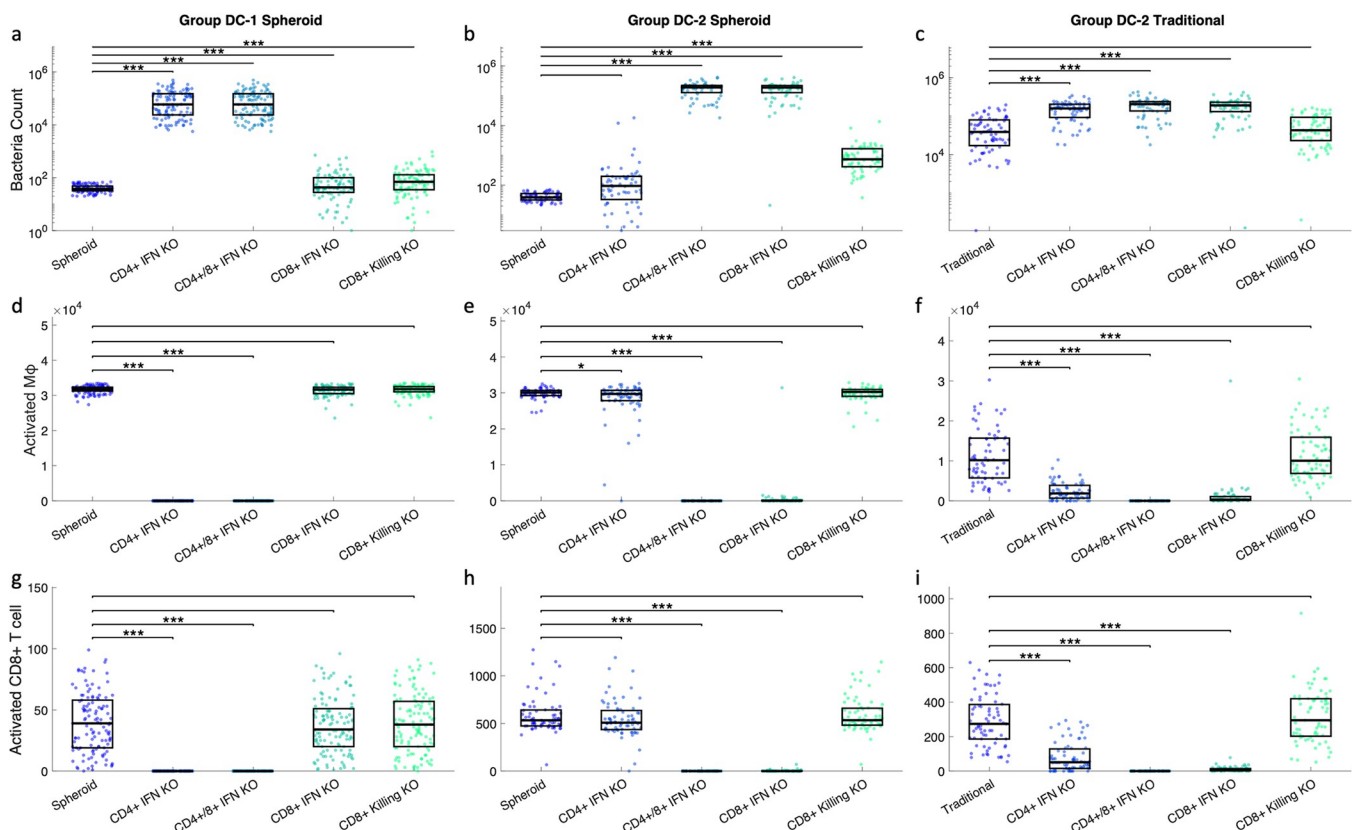

**Fig 8. Simulations were run with IFN-γ secretion virtual knockouts from CD4+ T cells, CD8+ T cells, or both.** A simulation knocking out CD8+ T cell killing of infected macrophages was also performed. The original simulation and the 4 knockout scenarios were compared by looking at a-c) bacterial counts at day 6, d-f) activated macrophage counts at day 4, and g-i) activated CD8+ T cell counts at day 4. These outputs are shown for the DC-1 spheroid simulations (n = 118), DC-2 spheroid simulations(n = 71), and DC-2 traditional simulations(n = 71).

setup than the spheroid, even though the DC-2 traditional simulations have the same parameter values as the DC-2 spheroid simulations. This indicates that, even while controlling for all parameters, changing the structure/movement not only changes bacterial control but also the mechanisms responsible for that control.

CD8+ T cells contribute to bacterial control both directly through cytotoxic killing and indirectly via IFN-γ production leading to macrophage activation. In DC-2 simulations, increases in both of these mechanisms are seen. KOs of CD8+ IFN-γ production and CD8 + cytotoxic killing were performed independently to see which contributes more to bacterial control. In DC-2 simulations, IFN-γ secretion from CD8+ T cells is more important in bacterial control than cytotoxic killing (Fig 8C). Some portion of macrophage and CD8+ T cell activation is dependent on this IFN-γ secretion (Fig 8F and 8I).

Taken together, our results predict a CD4+ T cell dominant response in DC-1 simulations and a CD8+ T cell dominant/mixed response in DC-2 simulations (Table 2). Note that DC-1 simulations are defined by lower CD4+ T cell doubling times, which are associated with increased CD4+ T cell secretions and lower CD8+ T cell generations due to the bimodal parameter distributions. CD8+ T cells are then compensating for reduced CD4+ T cell proliferation and secretion in DC-2 simulations. Not only is this seen in the responses to the KOs, but also in the absolute numbers of these cell types. Activated CD4+ T cells are present in larger numbers in DC-1 simulations (Fig 4I and Fig 6I), and the number of activated CD8+ T

**Table 2. Bacterial count relative to control for DC-1 and -2 spheroid and traditional virtual knockouts and the predicted dominant T cell fraction in these bacterial responses as based on the virtual knockout results.**

|  | DC-1 spheroid | DC-1 traditional | DC-2 spheroid | DC-2 traditional |
|---|---|---|---|---|
| CD4+ IFN KO | ↑↑ | ↑ | = | ↑ |
| CD4+/8+ IFN KO | ↑↑ | ↑ | ↑↑ | ↑ |
| CD8+ IFN KO | ↑ | = | ↑↑ | ↑ |
| CD8+ Killing KO | ↑ | = | ↑ | = |
| Dominant T cell | CD4+ | CD4+ | CD8+ | Mixed |

cells is larger in DC-2 simulations (Fig 4J nd Fig 6J). Our simulation therefore predicts multiple potential mechanisms of bacterial control between spheroid and traditional cultures: macrophage dominant in similarly controlled simulations, CD4+ dominant in DC-1 simulations, and mixed/CD8+ dominant in DC-2 simulations. This heterogeneity could be an indication of possible biological responses or could inform future data collection (e.g., CD4+ and CD8+ T cell counts at day 4) to further narrow parameter ranges and eliminate non-biological responses.

## 2.7 The initial structure and movement rules work synergistically to reduce bacterial load

We next aimed to answer whether the proximity of T cells and macrophages is predicted to be due to the initial structure of cells (i.e., layered spheroid vs randomly distributed traditional cultures) or due to the movement rules (i.e., free movement in 3D vs gravity-limited movement at the bottom of the simulation space). To separate these two factors, the movement rules for the simulations are swapped. The spheroid is simulated with gravity-limited movement rules (spheroid gravity), and the traditional setup is simulated with 3D movement rules (traditional floating). Surprisingly, both of these sets of simulations perform more poorly than the original setups in controlling bacteria (Fig 9A). Results are shown for DC-1 simulations only, but the DC-2 simulations show similar trends (S8 Fig). This implies that 3D or gravity-limited movement rules in and of themselves do not dictate the level of bacteria killing, rather they work along with the structure to reduce the bacterial load. To identify the mechanisms behind these impacts of cell movement and culture structure, the proportions of bacterial killing relative to the original sets are explored. The spheroid simulation with gravity-limited movement rules has almost no bacteria killed via activated macrophages and CD8+ cytotoxic cells when compared to the original spheroid (Fig 9B). This corresponds to a decrease in activation of macrophages, CD4+ and CD8+ T cells (Fig 9E-9G). The traditional simulations with floating have a smaller decrease in activated killing, but a larger decrease in CD8+ T cell killing (Fig 9B-9D) when compared to the traditional simulations. This corresponds with a small decrease in CD4+ T cell activation and a large decrease in CD8+ T cell activation (Fig 9F-9G).

Again, the spatial distribution of cells is analyzed by looking at mean distances between cells (Fig 9H and 9I). The macrophages are closest to activated T cells in the spheroid, followed by the traditional, floating traditional, and finally the spheroid gravity. This order corresponds to the number of activated macrophages in each simulation (Fig 9E). Macrophages being closer to activated T cells leads to more STAT1 activation, more overall activation, and more activated macrophage killing. Similarly, the distance from CD8+ T cells to STAT1 macrophages is smallest in the spheroid, followed by the traditional, traditional floating, and finally the spheroid with gravity (Fig 9I). Activated CD8+ T cell count in order of highest to lowest is traditional, spheroid, traditional floating, and spheroid gravity. So, proximity of CD8+ T cells

 Importance of in vitro tuberculosis granuloma structure explored with agent-based model

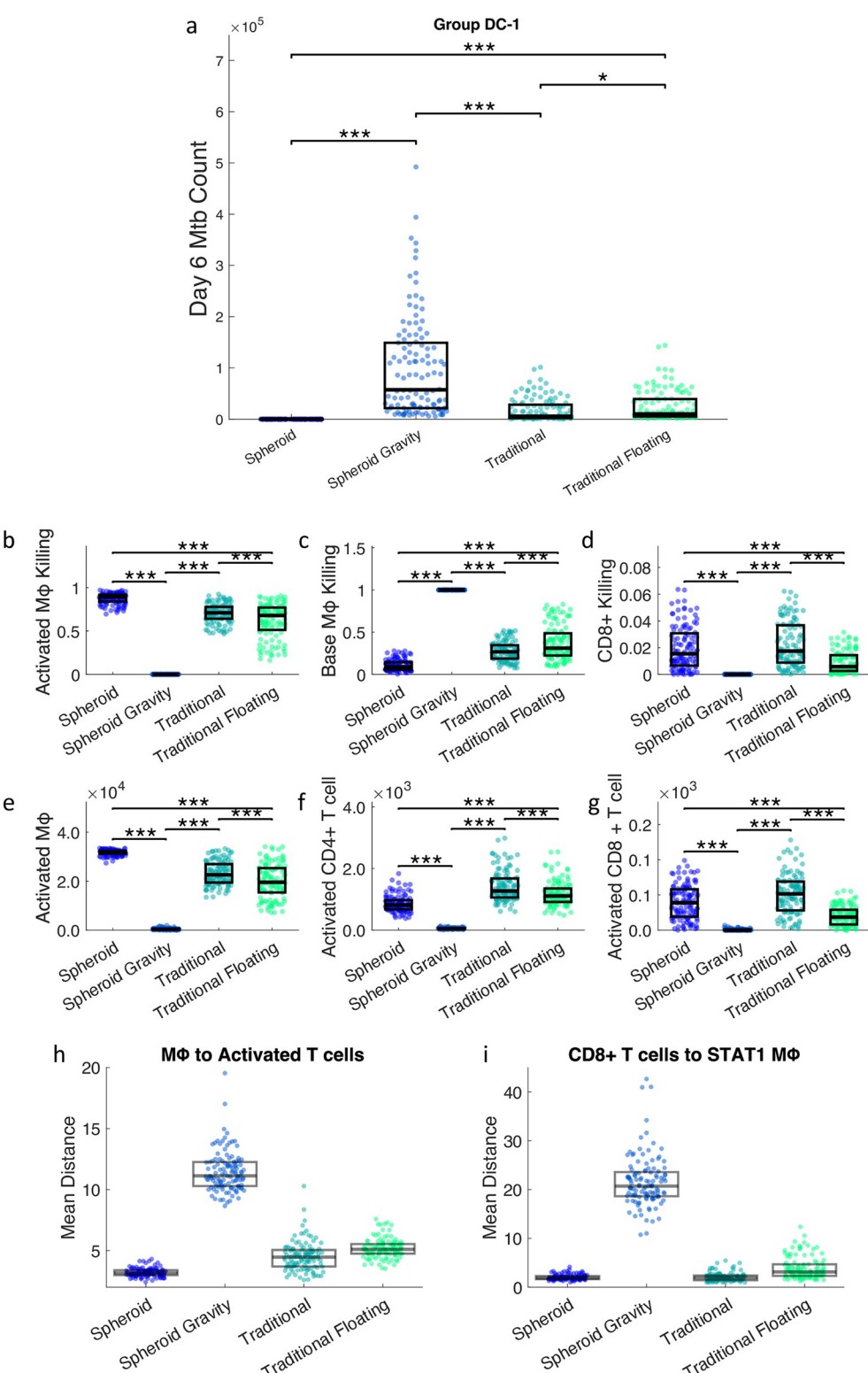

**Fig 9. The movement rules were swapped for the spheroid and traditional simulations, creating spheroid gravity and traditional floating.** a) Bacterial counts at day 6 in spheroid, spheroid gravity, traditional, and traditional floating scenarios. Proportions of bacterial killing due to b) activated macrophages, c) base macrophages, and d) CD8+ T cells from day 3 to 4. Total counts of e) activated macrophages, f) activated CD4+ T cells, and g) CD8+ T cells on day 4. h) Mean distance from macrophages to activated T cells on day 4. i) Mean distance from CD8+ T cells to STAT1

macrophages on day 4. This analysis is done with DC-1 simulations, the same analysis for DC-2 simulations is in the S8 Fig. * p< = 0.05, ** p< = 1e-2, *** p< = 1e-3.

to STAT1 macrophages corresponds with higher CD8+ T cell activation and more CD8+ T cell killing.

Traditional floating performs more poorly in terms of bacterial control, since it is hypothesized that the addition of the 3rd dimension as an option for movement effectively reduces the concentration of cells. The traditional floating cells are slightly farther apart than the equivalent cells in the traditional setup (Fig 9H and 9I), which is reflected in the macrophage and CD8+ T cell activation. The spheroid gravity also performs more poorly in terms of bacterial control. The spheroid gravity simulations start with the cells in similar initial conditions. But, as the movement rules are gravity-limited, the cells quickly fall to the bottom of the dish. As the macrophages are added at day 0 and the T cells are added at day 2, the macrophages form a compact dome that the T cells then surround. This conserves the layered structure, but it is forced to be denser. Due to the tight packing of the macrophages in this scenario, the T cells infiltrate less and have less opportunity for interaction (Fig 9H and 9I). This suggests that how the cells colocalize is dependent on both structure and density.

Swapping the movement rules showed that neither layered structure nor 3D movement independently improved bacterial control. Rather these changes predict reduced levels of macrophage and T cell activation, which corresponded to distances to activating cells. So, the layered structure and 3D movement of the cells are hypothesized to work together synergistically to activate immune cells and improve bacterial control.

## 3. Discussion

Our results suggest that differential control of bacteria in traditional and spheroid culture models is possible with only differences in the movement rules and initial cellular layout. Our model recreates multiple bacteria dynamics that are consistent with the experimental data, which predict multiple potential mechanisms of differential control. These mechanisms of differential control can be dominated by CD4+ or CD8+ T cells. The spatial layout of cells is predicted to be vital for improved immune response in the spheroid, as spatially localized signals and direct interactions between key cell populations are required for activation. Finally, we predict that bacterial control cannot be improved by 3D movement or layered structure independently.

Infection with *Mtb* can lead to disparate outcomes *in vivo* ranging from active disease to latent disease [40,41]. This variability exists at the host level and at the granuloma level as multiple granulomas within the same host can have widely different trajectories [42,43]. Similarly, our simulation shows a wide range of infection trajectories and parameter sets that can reproduce the experimental conditions. This variability in our simulations could represent differences between patients or granulomas, or just the stochasticity associated with infections. On the other hand, some of the variability in our model predictions could be due to parameter uncertainty. Further experiments with additional intermediate time points, especially day 3 and 4 when our simulations predict divergence between traditional and spheroid cultures, could help determine how much of this predicted variability is due to inherent biological variability and how much is due to uncertain parameter ranges. By looking at the simulation groups, we see bacterial counts and T cell activation, especially activated CD8+ T cells, differ the most between the three different qualitative behaviors (similarly controlled, DC-1 and DC-2). Therefore, we recommend that CFU and activated CD8+ T cells be measured at day 3. Considering that bacterial counts could in turn affect the degree of cell activation, these

outputs could also be considered in the context of different bacterial MOIs. Another differentiating experiment could be recreating the IFN-γ secretion KOs we simulated. Assessing both the spheroid and traditional culture outcome (CFU), the *Mtb* specific and non-specific T cell activation, and the relative dominance of CD4+ or CD8+ T cells in the context of conditional KOs would differentiate the groups. If we can get measurements for those metrics, we could possibly eliminate behaviors that are not biologically relevant. Nonetheless, our simulations identify new hypotheses for diverse possible responses to *Mtb* infection, that will inform future studies.

IFN-γ and downstream STAT1 have long been known to play a vital role in containing *Mtb* [44,45]. IFN-γ is used *in vitro* to polarize macrophages towards M1, or classical activation. This activation leads to increased autophagy, phagolysosome maturation, reactive oxygen species production, and bacterial control. Interruptions in IFN-γ production and its signaling pathway lead to disseminated TB in mice and cause Mendelian Susceptibility to Mycobacterial Diseases in humans [45,46]. Given IFN-γ's role in *Mtb* control *in vitro* and *in vivo*, it makes sense that this would be a predicted mechanism for improved bacterial control in these simulations. *In vitro* monolayer and spheroid cultures did have similar levels of IFN-γ across the course of the *in vitro* experiment [11]. However, our model predicts that the total amount of IFN-γ is not the only consideration in bacterial control, but also the spatial distribution of IFN-γ and IFN-γ secreting cells.

*In vitro* work has shown that IFN-γ is not limited to immune synapses, but can act on and activate bystander cells [47]. This means that while IFN-γ is secreted in a polarized manner toward target cells, it can still act on non-target cells 150 μm away, albeit with slower dynamics [47]. *In vivo* IFN-γ is known to bind to the extracellular matrix, which can further localize these signals [48]. IFN-γ signaling is mediated by the phosphating and activation of STAT1 [49], and the spatial pattern of STAT1 phosphorylation has been examined in the context of TB. In pleural biopsies, widespread STAT1 phosphorylation was seen in CD68+ cells [50], which are generally found in the core of organized granulomas [51]. STAT1 phosphorylated cells were also found localized closer to the center of NHP granulomas in the epithelioid macrophage region [18,52]. The localization of IFN-γ signaling to specific regions of the granuloma on the periphery of the core is consistent with our prediction of STAT1 signaling in the spheroid simulation. Biopsies from controlling and noncontrolling human or NHP granulomas, with subsequent spatial transcriptomic analysis, will provide valuable information testing the importance of cell and signal localization in *Mtb* control.

While CD4+ T cells are an essential component of protection in TB, CD8+ T cells also play a role in protection [53]. CD8+ T cell deficiency reduces survival in mice [54], and knockouts in many CD8+ T cell pathways (i.e., MHC class I, TAP1, or β2-microglobulin) show impaired TB control [55,56]. Vaccines are also able to elicit a response through CD8+ T cells in the absence of CD4+ T cells [56]. CD8+ T cells are thought to act in TB primarily through the release of IFN-γ rather than through cytolytic mechanisms [57,58]. Cytolytic mechanisms include the release of granules with perforin and granzyme and the interactions of Fas receptors and ligands. Perforin gene-knockout, granzyme gene-knockout, and Fas receptor-defective mice all showed similar TB disease progressions to control mice, suggesting that cytotoxicity is not the main mechanism of action for CD8+ T cells in TB [59,60]. Meanwhile, IFN-γ production was required by CD8+ T cells to provide protection against *Mtb* infection when transferring T cell populations in athymic mice [58]. Previous differential equation modeling has predicted both of these mechanisms act together to control infection [61]. One of the predicted groups in our model (DC-2) shows that higher CD8+ T cell activation plays a role in bacterial control. This contribution is predicted to be primarily due to IFN-γ

production rather than cytotoxicity, which aligns with this literature [57,58]. This dominance of IFN-γ secretion over cytotoxic function is independent of culture conditions for DC-2 simulations.

DC-1 and DC-2 simulations can be differentiated on what subset of T cells mediates control (CD4+ dominant or CD8+ dominant, respectively). This dichotomy could be an indication of possible biological responses or could reflect uncertainty in parameter estimates. However, a similar variation can be seen in mouse strains. When examining the response to a potential vaccine, BALB/c mice have a CD8+ dominant response, while C57BL/6 mice have a more balanced CD4+/CD8+ response [62]. This suggests the heterogeneity in our results could reflect biologically reasonable variation.

Our understanding of the diversity and complexity of spatial organization of granulomas is still evolving. Immunohistochemistry has been used to evaluate the layouts of cells and molecular signals [63]. Multiplex immunofluorescence was recently used to look at the distribution of macrophages, T cells, and B cells in tissue [64]. Individual cells were mapped permitting complex spatial analysis, allowing the defining of lesion types, and showing a diverse spectrum of lesion composition [64]. CD68+ cells were closest to the core of necrotic lesions, followed by CD8+ T cells. In combination with the peripheral location of STAT1 phosphorylation, this suggests colocalization of CD8+ T cells and STAT1 activated macrophages, mirroring our prediction. They also found that the spatial makeup of non-necrotizing lesions is influenced by the proximity of nearby necrotizing granulomas, suggesting that the spatial distribution of cells is important not only within a lesion but also between lesions. These methods could be used to do single-cell spatial analysis that could be compared to computational predictions in our work. Granulomas have also been dissected using laser-capture microdissection, mass spectrometry, and confocal microscopy [3]. They found spatial distinct areas of pro- and anti-inflammatory activity consistent with previous work looking at regions of pro-and anti-inflammatory macrophages in NHP and human granulomas [65], suggesting that the spatial layout of these signals is important to kill bacteria while limiting tissue destruction. The location of T cells has also been looked at using confocal microscopy of live NHP granuloma sections, showing poor penetration of T cells into the macrophage core [66]. Computational modeling predicts that the specificity and location of T cells within the granuloma impact macrophage-T cell interactions, activation of T cells, and thus bacterial control in NHP granulomas [2]. This work and our work suggest that the organization of signals plays an important role in immune response.

Immunohistochemistry and immunofluorescence are limited in that only 2D slices of the three-dimensional structures are investigated, so the entire structure might not be accurately represented. More recently, micro-computed tomography has been used to look at the three-dimensional structure of human granulomas *ex vivo* [67,68]. They have found that granulomas are heterogeneous, complex, and can have branching structures, challenging the spherical description of granulomas commonly seen in literature.

Potential next steps include analyzing spatial layouts of cells to confirm that *in vitro* models are similar to *in vivo* systems, using computational models to simulate various complex pathologies to see what role space has in bacterial control, and looking at more complex cell culture systems. We find that the spatial layout of cells is predicted to impact immune response and bacterial control *in vitro*. How this relates to *in vivo* Mtb dynamics and granulomas is less clear. Perhaps monolayers better recreate the layer of cells lining a cavity or initial infection in the airways, while spheroids more closely resemble smaller cellular aggregates or early granulomas. Our results suggest that structure can impact immune response and bacterial control in spheroid and traditional cultures, but there are more complex pathological structures that could similarly influence immune response and bacterial control. *In silico* models allow

structure to be controlled in a precise way. Structures, like those obtained from micro-CT, could be simulated to explore how more complex structures impact immune response and/or how much immune response leads to emergent structure formation. Lastly, our analysis has been focused on one cell culture system. Simulating more advanced cell culture systems, such as those that include extracellular matrix or co-culture with non-immune cells could give further insight into the impact of spatial layout, particularly in early events in *Mtb* infection.

Limitations of the established model were discussed in Petrucciani et al. [15]. Briefly, simplifications were made to the macrophage activation pathway and certain populations of T cells were excluded (i.e., regulatory and γδ). Another consideration is that asserting that differential bacterial control is solely due to dimension is a simplifying assumption. We include the impact of architecture of cells, cell-cell interactions, and diffusible molecules, but do not take into account other dimension-influenced mechanisms like mechanical forces and forced polarity. These mechanisms could be explicitly included in future work, which should be driven by experimental data. Additionally, the experimental work and subsequent computational predictions are limited to 6 days. Longitudinal experimental assessment will be important moving forward. Finally, this system is limited by the exclusion of lung tissue or stroma. While the spheroid response may favor mycobacterial control, we do not yet know if that response may be damaging in the context of lung tissue destruction. This destruction can be explored through the use of organoids or the inclusion of collagen, lung epithelial cells, and/or stromal cells. A balance in protection versus tissue integrity is required, for an overall successful response to infection.

This work is intended to be hypothesis generating, especially given the limited experimental calibration data. However, our model is also informed by and constrained by many immunological mechanisms and parameter ranges from literature. Further, we present and analyze multiple varying dynamics that were seen within the calibrated set, thereby taking into account model uncertainty. These model-predicted dynamics can be used to 1) help identify important data points and time points that should be measured in the future and 2) assist in generating and experimentally testing hypotheses that explain experimentally observed differences. Thus, this work guides experimental data collection, aids data analysis and interpretation, as well as guides and improves future iterations of computational and experimental feedback.

In summary, we use an *in silico* model that can represent multiple *in vitro* TB infection models to explore the impact of dimension and structure on bacterial control. Our results suggest that the spatial distribution of cells and mediators play an important role in immune activation and, therefore, a successful anti-mycobacterial response. This is especially important when direct interaction (CD8+ activation) or localized signals are necessary (IFN-γ secretion). This model structure is poised to simulate more *in vitro* systems and be extended to address further questions through the inclusion of collagen and non-immune cells.

## Supporting information

**S1 Table. Parameters descriptions for the parameters that are varied during calibration.** (DOCX)

**S2 Table. Parameters that were held constant during sampling, their values, and units.** (DOCX)

**S1 Fig. The distributions of parameters for similarly controlled simulations, differentially controlled simulations, DC-1 simulations, and DC-2 simulations.** Each parameter's range has been normalized from 0 to 1 based on the initial lower and upper bounds of the parameters

ranges as given in Table 1.
(TIF)

**S2 Fig.** a) Bacterial dynamics for similarly controlled simulations. The time range of interest (Day 3–4) is outlined. These are the same time courses as shown in Fig 3b. Proportions of bacterial killing due to b) activated macrophages, c) base macrophages, and d) CD8+ T cell killing from day 3 to 4. Comparing total counts of e) activated macrophages, f) STAT1 activated macrophages, g) NF-κB activated macrophages, h) activated T cells, i) activated CD4+ T cells, and j) activated CD8+ T cells between spheroid and traditional simulations at day 4.
(PDF)

**S3 Fig. Number of activated macrophages, STAT1 macrophages, NFκB macrophages, and T cells from day 2 to day 6 in DC-1 simulations. ** $p< = 1e-2$, *** $p< = 1e-3$.**
(TIF)

**S4 Fig.** The mean distance from macrophages to nearest a) T cells, b) activated T cells, c) activated CD4+ T cells, and activated CD8+ T cells. Each data point represents an average of all cells in a single simulation. The distribution of the mean distances across the set of simulations is compared between DC-1 spheroid and traditional simulations from day 3 to day 6.
(PDF)

**S5 Fig. Number of activated CD8+ T cells, STAT1 macrophages, and macrophages that have interacted with bacteria from day 2 to day 6 in DC-2 simulations.** * $p< = 1e-1$, ** $p< = 1e-2$, *** $p< = 1e-3$.
(PDF)

**S6 Fig.** The mean distance from a) TB-specific CD8+ T cells to nearest macrophage and b) TB-specific CD8+ T cells to nearest STAT1 activated macrophage. Each data point represents an average of all cells in a single simulation. The distribution of the mean distances across the set of simulations is compared between DC-2 spheroid and traditional simulations from day 3 to day 6. * $p< = 0.05$, *** $p< = 1e-3$, **** $p< = 1e-4$.
(PDF)

**S7 Fig. Simulations were run with IFN-γ secretion knockouts from CD4+ T cells, CD8+ T cells, or both.** A simulation knocking out CD8+ T cell killing of infected macrophages was also performed. The original simulation and the 4 KO scenarios were compared by looking at ab,I,j) bacterial counts at day 6, c,d,k,l) activated macrophage counts at day 4, e,f,m,n) activated CD4+ T cells counts at day 4, and g,h,o,p) activated CD8+ T cell counts at day 4. These outputs are shown for the DC-1 spheroid simulations (a,c,e,g), DC-2 spheroid simulations (b,d,f,h), DC-1 traditional simulations (I,k,m,o), and DC-2 traditional simulations (j,l,n,p).
(PDF)

**S8 Fig. DC-2 simulations movement rules swap experiment results.** Comparisons between the original setups (spheroid and traditional) and those with swapped movement rules (spheroid gravity and traditional floating) for *Mtb* count at day 6(a); proportion of bacterial killing due to activated macrophages (b), base macrophages(c), and CD8+ T cell cytotoxic killing of macrophages and internalized bacteria(d) from day 3–4; and levels of activation in macrophages(e), CD4+ T cells(f), and CD8+ T cells(g). * $p< = 0.05$, ** $p< = 1e-2$, *** $p< = 1e-3$.
(PDF)

**S9 Fig. Covariance of varied parameters for DC-1 and DC-2 simulations.** (p = 0.01 with Bonferroni correction).
(TIF)

## Acknowledgments

We thank Lev Gorenstein and the rest of the Research Computing Staff for their assistance with batch computing at the Rosen Center for Advanced Computing. We would also like to acknowledge Catherine Weathered for her mentorship and her work setting up the foundations in Repast and Slurm for our lab.

## Author Contributions

**Conceptualization:** Elsje Pienaar.

**Data curation:** Leigh Kotze.

**Software:** Alexa Petrucciani, Alexis Hoerter.

**Supervision:** Nelita Du Plessis.

**Visualization:** Alexa Petrucciani.

**Writing – original draft:** Alexa Petrucciani.

**Writing – review & editing:** Alexis Hoerter, Leigh Kotze, Nelita Du Plessis, Elsje Pienaar.

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
