## [Decision Letter · Decision Letter 0]

23 Jan 2024

Dear Dr Pienaar,

Thank you very much for submitting your manuscript "Agent-based model predicts that layered structure and 3D movement work synergistically to reduce bacterial load in 3D in vitro models of tuberculosis granuloma" for consideration at PLOS Computational Biology.

As with all papers reviewed by the journal, your manuscript was reviewed by members of the editorial board and by several independent reviewers. In light of the reviews (below this email), we would like to invite the resubmission of a significantly-revised version that takes into account the reviewers' comments.

Overall, the reviewers indicate a need for clarification of the modeling approach, the justifications for this approach, and the results. In particular, although the full model is explained in a separate preprint and the parameter ranges are tabled in the supplement, the reviews indicate that it would benefit this manuscript to include a brief description of the parameterization process and of the parameter distributions/constraints in the main text. Further, it should be made very clear early in the manuscript that this model is hypothesis generating, and the results should be presented more explicitly in that context.

We cannot make any decision about publication until we have seen the revised manuscript and your response to the reviewers' comments. Your revised manuscript is also likely to be sent to reviewers for further evaluation.

Sincerely,

Nic Vega, Ph.D.

Academic Editor

PLOS Computational Biology

Amber Smith

Section Editor

PLOS Computational Biology

Overall, the reviewers indicate a need for clarification of the modeling approach, the justifications for this approach, and the results. In particular, although the full model is explained in a separate preprint and the parameter ranges are tabled in the supplement, the reviews indicate that it would benefit this manuscript to include a brief description of the parameterization process and of the parameter distributions/constraints in the main text. Further, it should be made very clear early in the manuscript that this model is hypothesis generating, and the results should be presented more explicitly in that context.

Reviewer's Responses to Questions

**Comments to the Authors:**

Reviewer #1: The manuscript 'Agent-based model predicts that layered structure and 3D movement work synergistically to reduce bacterial load in 3D in vitro models of tuberculosis granuloma' by Petrucciani et al. aims at using two different topologies of an agent based model of tuberculosis granuloma. One topology corresponds to a 3D in-vitro model where cells can move freely in three directions and that is initialized with a core of macrophages and a shell of T cells, whereas the other corresponds to a 2D layer in-vitro model. The simulations contain macrophages, CD4 and CD8 T cells and bacteria.

Cells can secrete cytokines and move along chemokine gradients. Direct and cytokine mediated cell-cell contacts can change the activation status of cells.

The authors find different types of behavior in 3D and 2D simulations and discuss possible reasons.

The authors refer to the mechanisms described in their model by describing them as 'established' rules that have been previously published. 'This hybrid model was

parameterized in Petrucciani et al. with experimental data for bacterial fold change, cell count, and cell viability at day 6 from both the spheroid and the traditional cultures.' Since the cited publication has not been peer reviewed, it is unclear in what sense it represents an 'established' model.

Even though the manuscript does not describe the parameter variation process explicitly, Table 1 and its caption and some discussions in the manuscript suggest that parameters ranges are varied and values are drawn from certain distributions, with the structure of the latter remaining unclear.

In fact, many mechanisms and parameters remain unclear. For instance, what determines chemokine secretion? How are chemotactic gradients established? How do they depend on cellular states?

The aim of the manuscript is very laudable. Computational models offer the opportunity to analyze which mechanisms are responsible for certain features in experimental data. However, care must be taken to constrain model parameter values as tightly as possible based on experimental data. Here, the constraint is just one time point with cell and bacteria counts. For the amount of parameters that are varied, this is clearly insufficient, rendering the detailed discussion of the reasons behind the observed differential behavior of the model topologies rather moot, in my view.

Reviewer #2: This paper presents and explores the dynamics of an agent-based in silico TB infection model. The paper was well-organised and clearly written. Experimental data was incorporated to parameterise the model and used to compare against model behaviour. Results were clearly presented using appropriate statistics and the discussions of results were well-reasoned. The results and conclusion reached by the paper could have significant impact on furthering our understand of TB.

The references section needs careful proof reading and editing as it contains numerous errors: incomplete references (e.g. (5), (24)) and formatting errors (e.g. (10); (13); (22,23) - journal name inconsistent; (12) - full journal name).

Figure 1 p.9 - The caption of this figure needs more detail. In (B) the desciption doesn't describe the figure (e.g. CD3/PBMC are not mentioned in the text). Also, there's a mismatch between "B" and "b" (inconsistent use of upper/lower case).

There were a number of minor typographic errors which did not affect understanding but need correction:

* p.6 "An additional 10.6 people fell ill" - 10.6 people?

* throughout, remove contractions e.g. p.6 "hasn't"; p.25 "don't", p.41 "don't".

* throughout, references needs to be before full stops. e.g. p.6 "decades.(1)" -> "decades (1)."; The same comment applies to Figure and Appendix references.

* throughout, inconsistent use of bold font in Figure/Section references e.g. p.10 "Figure 1a" ("Figure 1" is in bold, "a" is not).

Reviewer #3: uploaded as attachment

**Have the authors made all data and (if applicable) computational code underlying the findings in their manuscript fully available?**

Reviewer #1: **No: **code access is not described in the manuscript.

Reviewer #2: Yes

Reviewer #3: Yes

PLOS authors have the option to publish the peer review history of their article (what does this mean?). If published, this will include your full peer review and any attached files.

Reviewer #1: No

Reviewer #2: **Yes: **Jamie Twycross

Reviewer #3: No
---

## [Decision Letter · Decision Letter 1]

9 May 2024

Dear Dr Pienaar,

Thank you very much for submitting your manuscript "Agent-based model predicts that layered structure and 3D movement work synergistically to reduce bacterial load in 3D in vitro models of tuberculosis granuloma" for consideration at PLOS Computational Biology. As with all papers reviewed by the journal, your manuscript was reviewed by members of the editorial board and by several independent reviewers. The reviewers appreciated the attention to an important topic. Based on the reviews, we are likely to accept this manuscript for publication, providing that you modify the manuscript according to the review recommendations.

The reviewers indicate that all substantial concerns have been addressed, but that there are a few minor points of clarity that should be addressed prior to publication. These points are entirely about presentation of the data and results and should be trivial to address.

Sincerely,

Nic Vega, Ph.D.

Academic Editor

PLOS Computational Biology

Amber Smith

Section Editor

PLOS Computational Biology

The reviewers indicate that all substantial concerns have been addressed, but that there are a few minor points of clarity that should be addressed prior to publication. These points are entirely about presentation of the data and results and should be trivial to address.

Reviewer's Responses to Questions

**Comments to the Authors:**

Reviewer #3: Excellent job addressing concerns. nice job!

Reviewer #4: This revised manuscript by Petrucciani et al. report a series of studies using their recently-described in vitro 3D spheroid granuloma model in order to better understand how specific permutations affect bacterial control. Tuberculous granulomas are complex structures with a network of interactions between multiple immune cell types and they are a key determinant of TB control or progression. Naturally formed granulomas from human or NHP are difficult to isolate and manipulate experimentally. In vitro models offer a tractable platform by which to test specific hypotheses. Here the authors describe a series of elegant simulations using their computational model recently described in PMID 38517920 to test hypotheses about the role of CD4+, CD8+, and macrophages in controlling Mtb within in vitro granulomas. The studies are well conceived and executed and the revised manuscript is very well written. Their in silico approach is hypothesis-generating and will guide the development of more complex in vitro models, inform in vitro and in vivo studies, and advance our understanding of the immune responses needed to protect from Mtb.

Reviewer #5: I agree with other reviewers in that this manuscript can be a nice contribution to the field. Although I agree with Reviewer 1 in that a very large list of parameters is being `calibrated' based on relatively few data points, the hypotheses and dynamics generated in this study (in silico) have the potential to be tested in the future in vitro/ex vivo/in vivo experiments. In the revised version of the manuscript, the main model hypotheses and used parameter calibration methods have been better explained.

Overall, I believe the authors have successfully addressed all comments from reviewers.

A few relatively minor points:

- I would recommend the authors to refer to Figure 2 in (15) where a nice overall description of the model is given.

- I believe that Table 1 is currently very much a reproduction of Table 1 in (15). I imagine there is no issue since the copyright of (15) is with the authors, and I can see the benefit of including this table in this manuscript. However, I would recommend making this clear and citing such table. Moreover, a list of parameters with values without a clear description of what each parameter represents is not necessarily useful, and makes the interpretation of results (corresponding to specific parameter values) difficult for the reader. Can the authors refer to a clear comprehensive glossary of parameter definitions from somewhere? Either from the Supplementary, the Github repository or Ref (15).

- Please make sure that resolution of all images is good enough.

- I would strongly recommend that key data is provided numerically, rather than just as figures (e.g. boxplots), for reproducibility purposes. Similarly, key summary statistics of calibrated parameters would facilitate reproducibility.

**Have the authors made all data and (if applicable) computational code underlying the findings in their manuscript fully available?**

Reviewer #3: Yes

Reviewer #4: Yes

Reviewer #5: **No: **I couldn't see numerical data provided (just figures). I might have missed it though.

PLOS authors have the option to publish the peer review history of their article (what does this mean?). If published, this will include your full peer review and any attached files.

Reviewer #3: No

Reviewer #4: No

Reviewer #5: No

Figure Files:

Data Requirements:

Reproducibility:

References:

---

## [Editor Report · Decision Letter 2]

21 Jun 2024

Dear Dr Pienaar,

We are pleased to inform you that your manuscript 'Agent-based model predicts that layered structure and 3D movement work synergistically to reduce bacterial load in 3D in vitro models of tuberculosis granuloma' has been provisionally accepted for publication in PLOS Computational Biology.

Best regards,

Nic Vega, Ph.D.

Academic Editor

PLOS Computational Biology

Amber Smith

Section Editor

PLOS Computational Biology

---

## [Editor Report · Acceptance letter]

10 Jul 2024

PCOMPBIOL-D-23-01717R2 

Agent-based model predicts that layered structure and 3D movement work synergistically to reduce bacterial load in 3D in vitro models of tuberculosis granuloma

Dear Dr Pienaar,

I am pleased to inform you that your manuscript has been formally accepted for publication in PLOS Computational Biology. Your manuscript is now with our production department and you will be notified of the publication date in due course.

With kind regards,

Olena Szabo
